# The Influence of Different Forms of Silver on Selected Pathogenic Bacteria

**DOI:** 10.3390/ma13102403

**Published:** 2020-05-23

**Authors:** Bogusław Buszewski, Agnieszka Rogowska, Viorica Railean-Plugaru, Michał Złoch, Justyna Walczak-Skierska, Paweł Pomastowski

**Affiliations:** 1Centre for Modern Interdisciplinary Technologies, Nicolaus Copernicus University in Torun, Wileńska 4, 87-100 Torun, Poland; bbusz@umk.pl (B.B.); 502538@doktorant.umk.pl (A.R.); rviorela@yahoo.com (V.R.-P.); michalzloch87@gmail.com (M.Z.); 2Department of Environmental Chemistry and Bioanalytics, Faculty of Chemistry, Nicolaus Copernicus University in Torun, Gagarina 7, 87-100 Torun, Poland; walczak-justyna@wp.pl

**Keywords:** silver nanocomposites, functionalization, kinetic, MALDI-TOF MS, bacteria molecular profile

## Abstract

The application of silver nanoparticles as an antibacterial agent is becoming more common. Unfortunately, their effect on microorganisms is still not fully understood. Therefore, this paper attempts to investigate the influence of silver ions, biologically synthesized silver nanoparticles and nanoparticles functionalized with antibiotics on molecular bacteria profiles. The initial stage of research was aimed at the mechanism determination involved in antibiotics sorption onto nanoparticles’ surface. For this purpose, the kinetics study was performed. Next, the functionalized formulations were characterized by Fourier transform infrared spectroscopy (FT-IR), dynamic light scattering (DLS) and a zeta potential study. The results reveal that functionalization is a complex process, but does not significantly affect the stability of biocolloids. Furthermore, the antimicrobial assays, in most cases, have shown no increases in antibacterial activity after nanoparticle functionalization, which suggests that the functionalization process does not always generate the improved antimicrobial effect. Finally, the matrix-assisted laser desorption/ionization time-of-flight mass spectrometry (MALDI-TOF MS) technique was employed to characterize the changes in the molecular profile of bacteria treated with various antibacterial agents. The recorded spectra proved many differences in bacterial lipids and proteins profiles compared to untreated cells. In addition, the statistical analysis of recorded spectra revealed the strain-dependent nature of stress factors on the molecular profile of microorganisms.

## 1. Introduction

In recent years, a significant increase in the number of bacterial strains resistant to commercial antibiotics has been observed. Nowadays, over 70% of bacterial infections are caused by strains resistant to at least one type of the most commonly used drugs [1]. The mechanism of microbial resistance is very complex, strain specific and can be influenced by many factors, such as growth conditions, life form or the metabolic state of the cells [2,3,4,5]. Therefore, the development of new substances with bactericidal properties is extremely important for health care [1,2,6,7]. Considerable interest is focused on the use of silver-based substances as an alternative to traditional drugs [1,2,8,9,10,11,12]. The antiseptic properties of silver have been known to mankind for centuries. Silver ions in the form of silver nitrate aqueous solution have experimentally proven antibacterial properties against a range of Gram-positive and Gram–negative bacteria [13]. Silver nitrate was probably used for the first time to treat local ulcers and chronic wounds [14]. With the development of nanotechnology, it became possible to synthesize silver nanoparticles with unique biological, chemical, and physical properties [2,12]. The antibacterial properties of silver nanoparticles against many microorganisms were confirmed in numerous studies [15,16,17,18,19,20,21]. Particularly attractive for modern medicine seem to be biologically synthesized silver nanoparticles, which, apart from their strong antibacterial properties, are non-toxic to eukaryotic cells and their synthesis is cheap, simple, and does not require hazardous chemicals [22,23,24,25,26,27,28,29]. Such methods include the use of microorganisms, fungi and plants, and the properties of nanoparticles obtained in this way can be easily designed by changing the pH, temperature or type of medium used during the process [9,25,26,27,28,29]. In the current study, silver nanocomposites synthesized by the probiotic Gram-positive bacteria strain *Lactococcus lactis*, were selected for research [9]. These studies also show that these nanoparticles are able to effectively inhibit the growth of many human pathogens such as *Pseudomonas aeruginosa*, *Proteus mirabilis*, *Staphylococcus epidermidis* or *Staphylococcus aureus*. Many recent studies also confirmed that the application of silver nanoparticles combined with antibiotics could result in the increase in their biological activity also in the case of antibiotic resistant strains [2,7,12,30,31]. Brown et al. [7] demonstrated that ampicillin functionalized silver nanoparticles were more effective in bacteria growth inhibition than non-functionalized counterparts against *Escherichia coli*, *Vibrio cholerae*, *Pseudomonas aeruginosa*, *Enterobacter aerogenes*, and *methicillin-resistant Staphylococcus aureus* (MRSA). Like in our previous studies, we confirmed that the functionalization of nanoparticles synthesized by the CGG11n *Actinomycetes* strain with tetracycline and ampicillin gave good results in increasing their antibacterial efficacy against many pathogens [10,12]. However, there are also some reports of the decrease in silver nanoparticles’ antibacterial properties after its combination with antibiotics. Jamaran et al. [32] noted the antagonistic effect of silver nanoparticles with gentamicin and neomycin against *S. aureus*. In addition, de Souza et al. [33] showed that the simultaneous application of amoxicillin and silver nanoparticles resulted in a decrease in their biological activity. For this reason, there is a clear need for further research to determine the mechanisms involved in the antibacterial action of such complexes and to explain the causes of their synergistic or antagonistic effects.

Although numerous studies confirmed the high potential of silver ions, silver nanoparticles and their complexes with antibiotics in the fight against microbes that are harmful to health, the exact mechanism of the interaction of such individuals with a bacterial cell remains unexplained [13]. Scientific reports indicate that antibacterial mechanisms of silver ions are associated with their interactions with the cell membrane [34], with molecules inside the cell such as enzymes, proteins (mainly S2 proteins of the small ribosome subunit) [35] or nucleic acids [36] as well as with the production of reactive oxygen species (ROS) [35]. In case of silver nanoparticles, many currently conducted studies suggest that the nature of their antibacterial properties is similar to silver ions [13]. The advantage of silver in the nano form is a very large surface to volume ratio, which results in a significant increase in antibacterial efficiency [37]. One of the main objectives of silver nanoparticles is the exterior barrier of the cell. They show the ability to adsorb on the surface of the cell wall and membrane [38]. Moreover, silver nanoparticles (AgNPs) interact with sulfur-containing cell wall proteins, leading to irreversible changes in its structure [39]. These changes have a significant impact on the integrity of the lipid bilayer and its permeability. An increase in cell membrane permeability results in the obstruction of proper intermembrane transport [36]. Furthermore, silver nanoparticles can interfere in biochemical pathways [13]. However, some results indicate that the bacteria are able to evolve resistance mechanisms to silver. Panáček et al. [40] found that some strains of *E. coli* and *P. aeruginosa* after a repeated exposure to silver nanoparticles could create resistance. Therefore, an alternative can be provided by the application of silver nanoparticles in the form of biocolloids as a support for antibiotics [41,42]. The effect of silver nanoparticles with the antibiotic particles adsorbed on its surface on the molecular profile of a bacterial cell can be significantly different as compared to the use of nanoparticles or antibiotics individually.

Taking into consideration the complexity of the action mode of silver-based antiseptic agents as well as their increasing application in many areas of human life, it is very important to conduct further research to clarify the silver–cell interactions. The matrix-assisted laser desorption/ionization time-of-flight mass spectrometry technique (MALDI-TOF MS), which allows insight into the molecular profile of the microbes, faces this challenge. This method is a new-generation tool enabling a thorough analysis of protein or the lipid profiles of microorganisms [43,44,45,46]. The signals of a given m/z value obtained on the recorded spectra constitute a unique “fingerprint” providing the possibility of specific, fast, and sensitive bacterial analysis [43,44]. Therefore, the application of this technology to track changes in the molecular profile of pathogenic bacteria treated with antibacterial agents can be crucial in explaining the mechanisms underlying their action. So far, several researchers have successfully used this technique to track changes in the metabolism of microorganisms exposed to various factors. Calvano et al. [44] applied this technique to investigate the lipids and proteins profile of *E. coli* treated with copper nanoparticles and ions. Rafińska et al. [8] reported that the MALDI technique is also useful in determining the impact of various forms of silver on the molecular profile of *Bacillus subtilis*. Moreover, our previous study on the influence of silver (bio) nanoparticles on probiotic bacteria and yeast cells’ metabolism confirmed the effectiveness of this technique to study changes occurring in cells [47].

The main goal of this study was to investigate the mechanisms involved in the antibacterial activity of different silver forms against various microorganisms on a molecular level. For this purpose, the MALDI-TOF MS technique was employed to investigate the effect on the bacteria molecular profile of silver ions, silver nanocomposites, antibiotics (tetracycline, ampicillin) and silver nanocomposites functionalized with antibiotics. The initial step of the study was to functionalize the LCLB56_AgNPs with ampicillin (AMP) and tetracycline (TET). In order to investigate the nature of antibiotics sorption on the surface of nanoparticles, the kinetic study was conducted and to confirm the occurrence of the functionalization the Fourier transform infrared spectroscopy (FT-IR) study was carried out. Moreover, the zeta potential value for nanocomposites before and after immobilization was measured. Next, the minimum inhibitory concentration (MIC) value for all tested compounds was determined for the common human pathogens *Escherichia coli* [48], *Staphylococcus aureus* [49,50], *Staphylococcus epidermidis* [51,52], *Klebsiella pneumoniae* [53,54,55] and *Pseudomonas aeruginosa* [56,57,58], and their antimicrobial potential was tested by flow cytometry. Finally, the changes in the proteins and lipids profile of bacteria after incubation with tested antiseptic factors was evaluated using the MALDI-TOF MS technique.

## 2. Materials and Methods

### 2.1. Chemicals and Reagents

Ampicillin and tetracycline as well as other chemicals, i.e., α-cyano-4-hydroxycinnamic acid, formic acid, solvents for high performance liquid chromatography (HPLC) and bacteria growth media (Mueller Hinton Broth), were provided by Sigma-Aldrich (Merck, St. Louis, MO, USA). Water was obtained from the Milli-Q RG system by Millipore (Millipore Intertech, Bedford, MA, USA).

### 2.2. Biological Material

Silver (bio) nanoparticles (LCLB56_AgNPs), used in this study for the functionalization step, were previously synthesized and characterized by Railean-Plugaru et al. [9].

The microorganisms used for the antimicrobial investigation assay *Escherichia coli* ATCC25922, *Pseudomonas aeruginosa* ATCC15442, *Klebsiella pneumoniae* ATCC BAA-1144, *Staphylococcus aureus* ATCC11632 were purchased from Sigma-Aldrich (Pol-aura, Zabrze, Poland) and *Staphylococcus epidermidis* from the collection of Centre for Modern Interdisciplinary Technologies, Nicolaus Copernicus.

### 2.3. LCLB56 (Bio) Silver Nanoparticle Functionalization Procedure

In order to prepare LCLB56_AgNPs functionalized with ampicillin and tetracycline, the equal volumes of each antibiotic solution (500 µg/mL) and silver (bio) nanoparticle suspension (700 µg/mL) were mixed and incubated with shaking at RT for 72 h. The concentrations of silver and antibiotics were chosen based on our previous study [10,59]. Next, obtained solutions were dialyzed using MWCO 3 500 Spectra/Por dialysis membrane (Spectrum laboratories Inc, Piscataway, NJ, USA) at RT for 72 h. CX 7500. The concentration of (bio) nanoparticles after functionalization was determined by ICP-MS Spectrometer. The samples once prepared were subject for future analysis.

### 2.4. Kinetics Study of Antibiotics Sorption onto LCLB56 Silver (Bio) Nanoparticles’ Surface

For this purpose, the samples were prepared in triplicate according to the Section 2.3. The antibiotic solution was diluted 1:1 (v:v), with water was used as a reference. The contents of the tube were mixed and then incubated at RT for 5, 10, 20, 40, 60, 300, 420, 720, 1080, 1800, 2160, 2520 or 2880 min. After incubations, the samples were centrifuged (RT, 15,000 rpm) and 300 µL of supernatant was transferred to the new Eppendorf tube. The concentration of ampicillin and tetracycline in obtained supernatants was determined by the HPLC (Shimadzu Prominence, Tokyo, Japan) analysis in accordance with the previously described method [10,12].

The amount of antibiotics bound by LCLB56_AgNPs (1), sorption effectiveness at each incubation period (2) and binding rate constants for zero (3), pseudo-first (4) and -second (5) order kinetic models as well as Weber–Morris intra-particle diffusion model (6) were calculated as it was described by Buszewski et al. [12], applying the following equations:q_t_ = (C_0_ − C) V/m(1)
E% = 100 × (C_0_ − C)/C_0_(2)
C = C_0_ − k_0_ × t(3)
q_t_ = q_e_ (1 − exp(−k_1_ × t))(4)
q_t_ = q_e_^2^ × k_2_ × t(1 + q_e_ × k_2_ × t)(5)
q_t_ = A + K_ip_ × t^0.5^(6)
where: q_t_—Amount of sorbed antibiotics at each time period (mg/g); m—Mass of sorbent (g); C_0_—Initial antibiotics concentration (mg/L); C—antibiotics concentration at each time period (mg/L); V—Sample volume (L); E—Sorption effectiveness (%); t—Time of sorption duration (min); k_0_—Rate constant of the zero-order kinetics model ((mg/L)/min); q_e_—Amount of antibiotics sorbed at equilibrium (mg/g); k_1_—Rate constant of the pseudo-first-order sorption kinetics model (min^−1^); k_2_—Rate constant of pseudo-second-order sorption kinetics model (g·mg^−1^ min^−1^); A—Constant that indicates the thickness of the boundary layer diffusion or external surface adsorption (mg/g); K_ip_—Diffusion rate constant ((mg/g)/t^0.5^).

### 2.5. Characteristics of Functionalized LCLB56_AgNPs

For the characterization of functionalized LCLB56_AgNPs, the Fourier transform infrared spectroscopy (FT-IR) analysis as well as the hydrodynamic size and the zeta potential measurements were conducted. In order to record the FT-IR spectra of ampicillin, tetracycline, LCLB56_AgNPs and LCLB56_AgNPs functionalized with antibiotics were mixed with of potassium bromide powder and the obtained samples were compacted. The spectra were recorded using the SPECTRUM 200 Perkin ELMER spectrophotometer (Perkin Elmer, Waltham, MA, USA) in the range of 400–4000 cm^−1^. In the case of the zeta potential and hydrodynamic size measurements, the Zetasizer NanoSeries and disposable folded capillary cells (Malvern Instruments, Malvern, UK) were used. The measurements of the nanocomposites’ zeta potential were conducted in different pH conditions. For this reason, directly before the analysis the samples were adjusted to the suitable pH. The zeta potential (*ζ*) was calculated according to the Smoluchowski equation:(7)$$\zeta =\frac{\mu_{e}\times \eta }{\varepsilon }$$
where: *μ_e_*—Electrophoretic mobility; *η*—Medium viscosity; *ε*—Dielectric constant.

For the hydrodynamic size measurements, the same procedure was applied. The dynamic light scattering (DLS) method was used for the size distribution of nanocomposites analysis. The measurements of zeta potential and hydrodynamic size of biocolloids was performed in triplicate for each of pH conditions.

### 2.6. Antimicrobial Potential

#### 2.6.1. Determination of the Minimum Inhibitory Concentration

The minimum inhibitory concentration (MIC) value for LCLB56_AgNPs functionalized with antibiotics as well as non-functionnalized LCLB56_AgNPs, ampicillin, tetracycline and silver ions against selected pathogenic bacteria were determined by the microdilution method in 96-multiwell microtiter plates. The bacterial culture and tested concentrations were prepared in the Mueller-Hinton (MH) broth media according to the Clinical and Laboratory Standards Institute (CLSI) methodology. Then, 125 μL of the antibacterial agent (final concentration of 200, 100, 50, 25, 12.5, 6.25, 3.125, 1.56 and 0.78 μg/mL; in case of functionalized silver (bio) nanoparticles, these concentrations meant the total amount of silver measured by inductively coupled plasma - mass spectrometry—ICP-MS) and 125 μL bacteria culture (final density of 1 × 10^6^ CFU/mL) were mixed and to each well 12 μL of in vitro toxicology assay kit, resazurin based (Sigma-Aldrich, St. Louis, MO, USA). After 24 h of incubation at 37 °C, the MIC value was determined visually based on the change in the redox indicator color from blue to pink or colorless. The lowest concentration at which no change in color was observed was considered as the MIC value, according to the Elshikh et al. (2016) [59].

#### 2.6.2. Flow Cytometry Analysis

The inoculation step for the flow cytometry analysis was prepared according to Section 2.6.1. based on the Clinical and Laboratory Standards Institute (CLSI) guidelines. The final concentration of tested samples (ampicillin, tetracycline, silver ions, LCLB56_AgNPs and functionalized LCLB56_AgNPs with antibiotics) for this assay was 25 μg/mL. Bacterial cells’ viability in the flow cytometry analysis was performed using a Live/Dead Double Staining kit (Merck, Darmstadt, Germany) as it was described by Ratiu and Railean-Plugaru et al. [60].

### 2.7. Molecular Changes on Bacterial Profiles as a Result of Antibacterial Factors Treatment Using MALDI-TOF MS Technique

#### 2.7.1. Bacteria Culturing

The samples were prepared as it was described in Section 2.6.2, with small modifications. At this step, based on the MIC value, the stress agents were added in such a way that the final concentrations in a sample constituted about 25% of the MIC value. The exception was *P. aeruginosa* with ampicillin, where the final ampicillin concentration was 100 μg/mL due to the bacterial resistance to this antibiotic. As a control, the bacteria cultured in pure growth media were used. Next, the prepared samples were mixed and incubated at 37 °C for 48 h. After this time, the culture was transferred to a sterile falcon tube and centrifuged (10 min, 10 °C, 4000 rpm). The obtained bacteria pellet was twice washed with sterile ultrapure water and subjected for the proteins (Section 2.7.2) and lipid investigation steps (Section 2.7.3). 

#### 2.7.2. Investigation of Protein Profile Changes of Selected Bacterial Cells

The protein extraction procedure from bacteria pellet (obtained as it was described in Section 2.7.1) and the MALDI-TOF MS analysis were performed according to the previously described methodology [47].

#### 2.7.3. Investigation of Lipids Profile Changes of Selected Bacterial Cells

The lipid extraction procedure consists of mixing of 1 mL of chloroform/methanol (2:1, v:v) with the obtained bacteria pellet in Section 2.7.1. The samples were then placed in an ultrasonic bath for 10 min, shaken for 20 min and centrifuged (RT, 10 min, 8000 rpm). The obtained supernatant was transferred to a new Eppendorf tube and washed with 200 µL of 0.05 M NaCl solution. The samples were mixed and centrifuged (RT, 10 min, 2000 rpm). The top water layer was removed and the bottom layer evaporated to dryness (37 °C). The sample obtained in such a way was resuspended in 20 µL of chloroform/methanol (2:1, v:v) mixture, mixed, sonicated and centrifuged (RT, 5 min, 2000 rpm). Next, 1 µL of each sample was spotted on the Ground Steel target in six repetitions and allowed to dry. The spectra were recorded in both reflective positive and negative ionization mode. For this reason, the α-cyano-4-hydroxycinnamic acid (HCCA) matrix (10 mg/mL) was prepared in the methanol or mixture of acetonitrile/water/trifluoroacetic acid (50:47.5:2.5, v:v:v) for a negative and positive ionization mode, respectively. A total of 1 µL of matrix in a suitable solvent was spotted on three spots and left to dry. The MALDI spectra were recorded at the 190–2500 m/z range on the Ultraflex Extreme II spectrometer with a smart beam laser (λ = 355 nm, 2 kHz frequency). Fragment spectra were recorded using the LIFT method. The FlexAnalysis and FlexControl software was used for the spectrometric data evaluation.

## 3. Results and Discussion

### 3.1. Kinetics Study of Antibiotics Sorption onto LCLB56 Silver (Bio) Nanoparticles’ Surface

To investigate the nature of the antibiotics sorption process on the surface of LCLB56 silver (bio) nanoparticles, a kinetic study was carried out. Figure 1A,B present the ampicillin and tetracycline sorption kinetic as a change plot of antibiotics concentration in solution per time unit, and Figure 1C,D summarized the antibiotics sorption effectiveness. It can be observed that, for both antibiotics, the sorption process is complex and non-linear but it has a slightly different nature depending on the antibiotic used. In the case of ampicillin, two different sorption steps can be identified. The first one is a very rapid sorption process (with k_0_ = 11.685 (mg/L)/min) that ends already after 10 min of antibiotic incubation with silver (bio) nanoparticles. During this time, about 97.08% ± 0.18% of ampicillin is bound and after that, the surface of the (bio) nanoparticles is saturated with the antibiotic, and the system reaches equilibrium. The maximum sorption capacity of (bio) nanoparticle amounts here to about 331.49 mg/g. The sorption process for tetracycline was quite different. In the case of this antibiotic, three distinct steps of the process are observed. The first one like previously is related to the rapid sorption process, but it is much longer and ends after 300 min when only the 19.26% ± 1.11% of the antibiotic is bound on the surface of silver (bio) nanoparticles. The second slower step is a gradual sorption step, during which 23.71% ± 1.91% of antibiotic was removed from the solution. This step ends after 1800 min of incubation and after that occurs a third step, which is a sorption equilibrium. The maximum sorption of silver (bio) nanoparticles was definitely lower for tetracycline in comparison to ampicillin and amounts to about 146.71 mg/g. 

The previous studies on the sorption of ampicillin [10] and tetracycline [12] by another type of biologically synthesized nanoparticles (synthetized by *Actinomycetes* strain CGG 11n) also showed that, in the case of respective nanoparticles, the mode of the sorption process for the studied antibiotics was different. Moreover, the sorption process seems to depend on the used antibiotic and to be similar for different types of (bio) nanoparticles. In the case of both types of (bio) nanoparticles, for ampicillin, only two steps can be identified and for tetracycline three steps were distinguished. However, the sorption capacity of LCLB56_AgNPs was much higher in the case of both studied antibiotics as compared to CGG 11n AgNPs (64.02 and 40.70 mg/g for ampicillin and tetracycline, respectively). The much higher efficiency of ampicillin sorption compared to tetracycline by different types of silver (bio) nanoparticles may result from differences in the elemental composition of both antibiotics. One of the explanations for this phenomenon may be that unlike tetracycline, the ampicillin molecule contains a sulfur atom. Therefore, it can be assumed that the rapid and highly effective sorption of ampicillin on the surface of silver (bio) nanoparticles is the result of the high affinity of the sulfur atom to silver (Figure 2) [61]. The mass, structure and chemical composition of the antibiotics may also correspond to their different sorption efficiencies [62]. The molecular mass of tetracycline amounts to 444.4 g/mol, while, for ampicillin, this value amount to 349.4 g/mol. In addition, tetracycline contains more rings in its structure. The smaller size of ampicillin molecules may therefore promote more efficient sorption, since more molecules of this antibiotic are needed to cover the surface of the (bio) nanoparticles. 

To calculate the rate constants of antibiotics’ sorption kinetic for the linear segments of the process the zero-order kinetics model was used. The application of the zero-order kinetic model is suitable for the description of sorption steps with a linear relationship. The units of the rate constant obtained by this model are the real physical parameter, which characterizes the sorption speed. The values of calculated rate constants were presented in Table 1. For ampicillin, the rate constant obtained for the first step amounts to 11.685 (mg/L)/min. In turn, for tetracycline, the rate constant amounts to 0.143 and 0.035 (mg/L)/min for first and second step, respectively. The calculated values clearly indicate that silver (bio) nanoparticle functionalization with ampicillin is definitely a faster process than its functionalization with tetracycline.

In order to present the accuracy of the obtained experimental data, the pseudo-first and -second-order kinetics models were applied. Figure 1A,B show the matching of experimental data to the pseudo-first and -second-order kinetics models, and Table 1 presents the calculated kinetics constants. The analysis of the calculated values of the relative approximate error (A_approx._) indicates that the pseudo-second-order kinetic model was more appropriate for the description of both the antibiotics’ sorption process on the surface of LCLB56_AgNPs. Furthermore, it can be observed that both of the applied models more accurately describe the sorption kinetics of ampicillin.

In order to better explain the mechanism occurring during the antibiotics sorption by LCLB56_AgNPs, the obtained results were tested against the Weber–Morris intra-particle diffusion model. Figure 1C,D show the plot of the functional dependence of the antibiotics sorption capacity of LCLB56_AgNPs on t^0.5^ (square root of time [min]). The results obtained for ampicillin reveal that, in the case of this antibiotic, the sorption mechanism relies only on the surface binding of the antibiotic and that there is no diffusion inside the (bio) nanoparticle, as evidenced by the presence of only two steps of sorption. In turn, for tetracycline, three steps can be distinguished. The first step is related to the surface sorption, the second one to intra-particle diffusion of the antibiotic inside of the silver (bio) nanoparticles structure and the final third step responds for the sorption equilibrium. The *y*-axis intercept of the line of the second sorption step allows for measuring the thickness of the external surface sorption and the slope of this line determines the value of the intra-particle diffusion coefficient obtained for the antibiotics sorption by LCLB56_AgNPs (Table 1). Moreover, the calculated values of the Gibbs free energy (ΔG^0^) change and the distribution coefficient (K_d_) of the antibiotics’ sorption onto LCLB56_AgNPs were −26.578, −16.564 kJ/mol and 50835.085, 857.123 for ampicillin and tetracycline, respectively (Table 2). These negative values of the Gibbs free energy indicate that in the case of both tested antibiotics the sorption onto LCLB56_AgNPs is a spontaneous process [12,63]. Moreover, a lower value of the Gibbs free energy obtained for ampicillin indicates greater process spontaneity, which is consistent with the kinetic data. In addition, the values obtained previously for CGG 11n AgNPs were definitely higher and amounted to −5.210 [12] and −12.377 [10] kJ/mol for tetracycline and ampicillin, respectively. However, despite the differences in the obtained values, the nature of the process was still spontaneous, and the value obtained for ampicillin was lower than for the second antibiotic.

### 3.2. Characterization of Functionalized LCLB56 Silver (Bio) Nanoparticles

To confirm the silver (bio) nanoparticle functionalization, the Fourier transform infrared spectroscopy approach was applied. Figure 3 presents the spectra recorded in the range of 400–4000 cm^−1^ obtained for LCLB56_AgNPs, tetracycline (TET), LCLB56 AgNPs functionalized with tetracycline (LCLB56_AgNPs/TET), ampicillin (AMP) and LCLB56_AgNPs functionalized with ampicillin (LCLB56_AgNPs/AMP). It was observed that the functionalization of nanocomposites with antibiotics led to changes in some vibration bands on the recorded spectra. These changes indicate the occurrence of the antibiotics sorption process and may provide the potential binding sites [12]. After nanocomposites’ functionalization with both antibiotics, the signals shifts from 2901 cm^−1^ (AMP), 2903 cm^−1^ (TET), 2908 cm^−1^ (LCLB56_AgNPs) to 2919 cm^−1^ (LCLB56_AgNPs/AMP; LCLB56_AgNPs/TET) and from 2869 cm^−1^ (LCLB56_AgNPs), 2865 cm^−1^ (TET) to 2851 cm^−1^ (LCLB56_AgNPs/AMP; LCLB56_AgNPs/TET) were noted. These changes are related with the presence of υ_s_(CH_3_) and υ_as_(CH_3_), respectively [64,65,66].

In addition, the signal shift from 1367 cm^−1^ (LCLB56_AgNPs), 1364 cm^−1^ (AMP) and 1357 cm^−1^ (TET) to 1357 cm^−1^ (LCLB56_AgNPs/AMP) and to 1358 cm^−1^ (LCLB56_AgNPs/TET), indicates the presence of υ(CCph) and υ(CO) antibiotic groups vibrations, respectively [65,66]. The signal shift from 953 to 960 cm^−1^ can be attributed to δ(CNC) and υ(CN) vibration from the ampicillin and tetracycline molecule, respectively. Moreover, the shift of the signal at 2842 to 2839 cm^−1^ is probably related to υ(CH_3_) vibrations in the case of both antibiotics [64,65,66]. Functionalization with antibiotics also leads to the appearance of a new signal at 1115 cm^−1^, which is probably derived from δ(HCN) and υ(CN) vibrations of the lactam ring of ampicillin and the tetracycline molecule, respectively. This suggests that these functional groups of antibiotics play a dominant role in their interaction with the surface of nanocomposites (Figure 2). Signal merging in the range of 2397–2302 cm^−1^ and a loss of signal at 2263 cm^−1^ after antibiotics sorption can be also noted.

Biologically synthesized LCLB56 silver nanoparticles contain a metallic core surrounded with organic coat composed mainly of proteins but also of lipids. Characteristics of the organic coat of (bio) nanoparticles by FTIR and MALDI techniques in our previous study showed that it consisted mostly amino acids such as histidine (His), arginine (Arg), tryptophan (Trp), tyrosine (Tyr) and aspartic (Asp) and glutamic (Glu) acid [9,67]. Therefore, signals present on the spectra of LCLB56_AgNPs at 1499, 1229 and 1042 cm^−1^ can be attributed to Tyr-O^−^ υ(CC), HisH δ(CH) and Glu-COO^−^ υ(CC) vibrations [68,69]. After silver (bio) nanoparticle functionalization with both antibiotics, these signals disappeared, which can indicate that these functional groups of the amino acids, which are present in the external organic layer of silver (bio) nanoparticles, participate in antibiotic binding (Figure 2). 

In order to investigate the dispersion stability of silver (bio) nanoparticles before and after functionalization, in different pH conditions, the zeta potential measurements were performed. The measurements were carried out over a wide pH range (2–11) to determine the possible application of nanocomposites. The pH values in the human body range from very acidic in the stomach (1.5–3.5) to slightly alkaline in the terminal ileum [70]. The Figure 4 shows the change plot of the zeta potential value in dependence on pH for all tested individuals. It was observed that functionalization did not significantly affect the value of the biocolloids zeta potential in given pH conditions. Moreover, it was noted that both functionalized and non-functionalized silver nanocomposites are non-stable at low pH conditions, as evidenced by the zeta potential value. The stability of the (bio) colloidal system can be easily predicted as based on the zeta potential value and the stability of dispersion ensures the value above of ±20 mV [12]. Based on the obtained results, it can be concluded that all types of tested silver nanocomposites (biocolloids) exhibit relatively high dispersion stability when the pH value exceeded 6. In addition, all the samples showed the highest stability at pH 7. In this exhibited condition, the zeta potential value amounts were found to be −34.60 ± 2.17 (B), −34.67 ± 0.95 (C) and −35.93 ± 1.74 (D) mV, for LCLB56_AgNPs, LCLB56_AgNPs/TET and LCLB56_AgNPs/AMP, respectively. Based on the obtained results, it can be assumed that tested biocolloids can be used to treat skin infections. In general, the pH of skin amounts to about 5.5. In addition, the pH of chronic wounds was found to be in the range of 5.45 to 8.65. Bacterial skin infections caused by strains such as *Klebsiella* spp, *P. mirabilis* or *P. aeruginosa* also lead to an increase in the skin pH. It is related with the production of urease by these strains [71]. Thus, biocolloids stable at a pH range of 6 to 11 can be effectively applied in the treatment of a wide range of diseases caused by bacterial infection.

The effect of functionalization on the size of silver nanocomposites was also investigated. Our previous research conducted by Transmission Electron Microscopy (TEM) technique showed that the tested silver (bio) nanoparticles were characterized by spherical shapes with the size of 5–50 nm [9]. However, the hydrodynamic size of (bio) nanoparticles, measured by dynamic light scattering (DLS), takes into account the hydration layer surrounding the biocolloid. Moreover, a component influencing the DLS size is brunched a solvated organic coat, naturally presented into/onto silver nanoparticles core. The hydrodynamic diameter is a key parameter, which characterizes the colloidal particles because it reflects their size in the aqueous solution and includes coatings or modifications performed on its surface. Therefore, the hydrodynamic size provides an indicator of the apparent size of the solvated particle and it can change due to the surface modification of the particle (e.g., by functionalization or agglomeration) [72]. To characterize the functionalized silver (bio) nanoparticles, the changes in the hydrodynamic size distribution profile at different pH conditions were investigated. Figure 5 presents the hydrodynamic size distribution (A) and intensity (B) of the recorded populations for biocolloidal systems of LCLB56_AgNPs and functionalized LCLB56_AgNPs/TET and LCLB56_AgNPs/AMP. In the case of non-functionalized LCLB56_AgNPs, three populations with different hydrodynamic size distributions were recorded (Figure 5A). One population presented the hydrodynamic size of nanocomposites in range of 90–260 nm (I) for all of the investigated pH, except pH 5 and 6, where the hydrodynamic size distribution slightly increased to the 860 and 450 nm, respectively. Moreover, the presence of a second, much smaller population was observed (II) with a hydrodynamic size value about 4500 nm in range of pH 2–4 and above 5000 nm in range of pH 7–9. Exceptions in the case of pH 6 and 10 were noticed. The hydrodynamic size values were found to be 90 (pH 6) and 136 nm (pH 10). In addition, the appearance of a third population (III) was observed only at pH 6 and 10 that could be related to the increase in the heterogeneity of the size of the main population of (bio) nanoparticles and a shift towards higher values: 5260 (pH 10) and 5560 nm (pH 6). However, the II and III population presented a very low intensity of the signal, meaning that those populations are insignificant when compared to the first population that have been found to be predominant (Figure 5B). The presence of populations with such large hydrodynamic size distribution is associated with a slight agglomeration of particles, and the generated size is the hydrodynamic diameter of several colloidal particles located close together in the solution. Considering LCLB56_AgNPs, the only one dominant population (I), with a higher intensity of the hydrodynamic size distribution (>90%) (Figure 5B) was noticed; it presented the lower size value (<260 nm), for almost all pH (2–9) except pH 10 (65%). The other observed populations (II and III) with the higher size distribution (>4000 nm) presented the hydrodynamic size distribution intensity of less than 5% (Figure 5B).

A similar phenomenon occurred in the case of functionalized LCLB56_AgNPs/AMP and LCLB56_AgNPs/TET. In both cases, under the analyzed range of pH, the hydrodynamic size distribution intensity of the I dominant populations was found to be more than 90% in contrast to other populations (II and III) where the hydrodynamic size distribution intensity was found to be less than 4% (Figure 5B). Moreover, considering LCLB56_AgNPs/AMP (Figure 5A), based on the hydrodynamic size distribution results, only two populations (I and II) were noticed with the average size as follows: I distribution with the average hydrodynamic size range about 600–900 nm for pH 2–6 and 140–190 nm for pH 7–10, II distribution with the average hydrodynamic size about 5300 nm for pH 6, 7 and 9. Contrastingly, in the case of LCLB56_AgNPs/TET (Figure 5A), three distributions (I, II, and III) were recorded, similar to non-functionalized LCLB56_AgNPs. All the recorded populations presented the following hydrodynamic size distribution values: I distributions—400–900 nm (pH 2–5), 140–200 nm (pH 6–10); II distributions –5200–5400 nm (pH 3, 5, 7–10), 114 nm (pH 2) and 40 nm (pH 6); III distribution—5100–5300 nm (pH 2, 6).

The functionalized LCLB56_AgNPs/AMP and LCLB56_AgNPs/TET systems are characterized by a slight increase in the size distribution value. The increase in the large size may be associated with the adsorption of a thick layer of the antibiotic on their surface, which is confirmed by kinetic studies indicating the rapid adsorption of significant amounts of the antibiotic and the absence of the intramolecular diffusion. In the case of those immobilized with the tetracycline hydrodynamic size distribution profile seems to be more similar to non-functionalized counterparts. This is because tetracycline is adsorbed in much smaller amounts than ampicillin. In addition, according to adsorption kinetics studies, part of the sorbed antibiotic is diffused into the interior of (bio) nanoparticles. However, the functionalization of LCLB56_AgNPs with this antibiotic also led to an increase in the size and greater uniformity of the main population [10,12]. Moreover, all the analyzed systems (LCLB56_AgNPs, LCLB56_AgNPs/AMP and LCLB56_AgNPs/TET) were still found to present the same stability, according to the zeta potential value.

### 3.3. Determination of Minimum Inhibitory Concentration and Flow Cytometry Analysis Antimicrobial Potential

The antimicrobial activity of LCLB56_AgNPs functionalized with tetracycline and ampicillin as well as non-functionalized LCLB56_AgNPs, silver ions, AMP and TET was tested against *E. coli*, *S. aureus*, *S. epidermidis*, *K. pneumoniae*, and *P. aeruginosa*, which are well known as one of the most common human pathogens. To this purpose, the minimum inhibitory concentrations (MIC) were determined and the flow cytometry analysis was applied. The determined MIC values for all variants are summarized in Table 3. The MIC study allowed us to compare the antimicrobial properties of different antimicrobial agents. All tested bacteria strains were already sensitive to tetracycline at low concentrations. Silver ions also showed very strong antibacterial effectiveness. In turn, ampicillin was definitely less effective. Satisfactory results were obtained only for *E. coli*, *S. epidermidis* and *S. aureus*, while the MIC value for *K. pneumoniae* amounted to 50 µg/mL and *P. aeruginosa* were completely resistant to this antibiotic which is in line with the data provided by the European Committee on Antimicrobial Susceptibility Testing (EUCAST). Silver (bio) nanoparticles showed good antibacterial properties and the MIC values for all tested bacteria strains were in the range from 6.25 µg/mL for *E. coli* to 12.5 µg/mL for *S. aureus*, *S. epidermidis*, *K. pneumoniae*, and *P. aeruginosa*. The functionalization of silver (bio) nanoparticles with tetracycline did not increase their antibacterial properties. Moreover, in the case of *E. coli*, the determined MIC value was even higher than for non-functionalized counterparts. Similarly, in the case of (bio) nanoparticles functionalized with ampicillin, no increase in the microbial activity against *P. aeruginosa* was also observed. In the case of other strains, a two-fold increase in the MIC value was observed as compared to unmodified (bio) nanoparticles. The exception was the use of this complex against *S. epidermidis*. Only in this case was the observed increase in biological activity as a result of functionalization.

In order to further explain the effectiveness of the tested substances, the flow cytometry analysis was performed. The studies were conducted under the same conditions for all of the investigated samples using the effective concentrations associated with the application. Taking into consideration the higher MIC values of LCLB56_AgNPs/AMP against both *K. pneumoniae* (25 µg/mL) and *S. aureus* (50 µg/mL), the effectiveness of tested agents was evaluated at concentrations of 25 µg/mL; in order to cover most cases. Figure 6A presents the percent of the total cell number, which grew after the 24-h incubation of bacteria with ampicillin, tetracycline, silver ions, and LCLB56_AgNPs functionalized and non-functionalized with antibiotics as well as without any stress agent (control), while Figure 6B–D show examples of cells’ fluorescence after their incubation with the given stress agent. It can be observed that, in the case of silver ions, a higher number of cells was observed than in the case of LCLB56_AgNPs. This is associated with the presence of both live and dead cells (Figure 6A). Therefore, it can be assumed that both silver ions and (bio) nanoparticles showed a good antibacterial activity but based on a different mode of action. Silver ions caused cell death, while silver (bio) nanoparticles inhibited their growth. In addition, after the application of (bio) nanoparticles at the obtained fluorescence spectra signals’ characteristic for disrupted cells can be observed (Figure 6C). It is well known that silver ions have strong antibacterial properties, but they can be also toxic for eukaryotic cells [73]. In turn, according to the previous study described by Railean-Plugaru et al. [9], LCLB56_AgNPs at lethal concentrations for bacteria do not affect the viability of fibroblast cell line L929. It may be related to the fact that the antibacterial action of silver (bio) nanoparticles is not only associated with the release of free ions, and the mechanism involved in their activity is much more complex. It is believed that one of the causes of the antibacterial action of nanoparticles is damage of the bacterial membrane, which in turn leads to cell lysis [1,74]. This mechanism of silver nanoparticle action can be confirmed by our analysis of the flow cytometry, where the incubation of bacterial cells with silver (bio) nanoparticles generated the appearance of the fragmented cells.

The results obtained for ampicillin brought similar results to the MIC studies that this antibiotic was not effective against *K. pneumoniae* and *P. aeruginosa*. As it was observed on the fluorescence spectra of bacteria after the incubation of these bacteria with ampicillin, all of the cells were alive (Figure 6B). In contrast, ampicillin combined with silver (bio) nanoparticles successfully inhibited the growth of these bacteria. Other results indicate that both unmodified and modified silver (bio) nanoparticles exhibit good antibacterial activity in relation to all tested bacterial strains. However, no differences have been observed in the antimicrobial potential values of non-functionalized and functionalized silver (bio) nanoparticles with antibiotics. It seems that the antimicrobial effect has been generated by the silver (bio) nanoparticles. Fluorescence spectra also showed that the incubation of (bio) nanoparticles with bacteria led to the disruption of cells into fragments (Figure 6C). The exception was (bio) nanoparticles immobilized with ampicillin against *S. aureus*. In this case, no inhibition of bacterial growth was observed, which may suggest antagonistic effects of (bio) nanoparticles and this antibiotic. However, more detailed studies showed that, in this case, next to the signal from live cells, which constitute 62% of all cells, there is also a signal from dead cells (38%) (Figure 6D). This indicates that this complex elicits microbial activity against the tested strain.

Many previous studies confirmed the increasing antibacterial efficiency of silver nanoparticles and antibiotics. Shruthi et al. [2] showed that different types of silver nanoparticles after its functionalization with streptomycin had better antibacterial activity than non-functionalized nanoparticles against *S. aureus* and *E. coli*. In addition, Brown et al. [7] demonstrated that chemically synthetized silver nanoparticles functionalized with ampicillin inhibit the growth of bacteria resistant to ampicillin more effectively. Moreover, our previous studies of CGG11n AgNPs indicate that its functionalization with ampicillin [10] and tetracycline [12] for most of the tested strains resulted in an increase in their antibacterial properties. In the current study, an increase in the biological activity as a result of functionalization was observed only when using ampicillin-functionalized (bio) nanoparticles against *S. epidermidis*. In other cases, the antibacterial ability of nanocomposites did not change after functionalization or even slightly decreased. It is noteworthy that the non-changes in antimicrobial activity after functionalization is equivalent with the antimicrobial potential generated by the silver nanocomposites that present high effectiveness even at low concentrations. On the other hand, the decreases in antibacterial properties of functionalized silver nanoparticles can be related with the formation of a thick layer of antibiotic on the surface of (bio) nanoparticles, which could limit the interaction of the nanoparticle with the bacterial cell surface. Such a phenomenon was also observed against *S. aureus* in our previous study investigating functionalized silver (bio) nanoparticles/AMP, synthetized by CGG11n *Actinomycetes* strain [10]. The decrease in the antibacterial properties of silver nanoparticles in combination with antibiotics (gentamycin and neomycin) against one of the tested *S. aureus* strain was also reported by Jamaran et al. [32]. Another study that confirms the possibility of antibacterial efficiency reduction of antibiotics and silver nanoparticles was the work of de Souza et al. [33], where the combination of AgNPs with amoxicillin results in the reduction of its antibacterial effect against *S. aureus*. In present study, the formation of a thick layer of antibiotic on the surface of (bio) nanoparticles was mainly seen in the case of ampicillin functionalized (bio) nanoparticles, where kinetics and hydrodynamic size distribution studies confirmed the formation of a thick surface layer. Since the antibacterial action of silver nanoparticles involves the release of free silver ions, which then interact with the bacterial cell and lead to its death, the presence of an antibiotic layer on the surface of (bio) nanoparticles can significantly reduce the amount of released free ions [74,75]. According to Khatoon et al. [76], amine groups of ampicillin can effectively reduce the silver ions to nanoparticles and then physically adsorb on their surface.

### 3.4. Molecular Changes on Bacterial Profiles as a Results of Antibacterial Factors’ Treatment Using MALDI-TOF MS Technique

The exact mechanism of antibacterial activity of silver ions, silver nanoparticles as well as their combinations with commercially available antibiotics still remains unclear. Many potential causes of antibacterial action of silver nanoparticles on bacterial cells were suggested [1,41,74,75]. The current research indicates that the antimicrobial properties of silver nanoparticles result in their ability to release free silver ions [77,78], antibiofilm activity [21,79] as well as its direct contact with the bacteria membrane, resulting in its damage [80,81]. It is also well known that silver ions may induce oxidative stress in microbial cells [8,13,82]. Moreover, in the case of antibiotic functionalized silver nanoparticles, it is supposed that the increase in the biological activity is associated with the role of nanoparticles as a drug carrier as well as a different mode of action of both agents [10,41,83]. However, in the present study, no increasing in the antimicrobial effect has been observed in case of functionalized compared to non-functionalized silver (bio) nanoparticles. Therefore, to investigate the effect of functionalization on the changes occurring at molecular level, the further research was conducted. Due to the increasing use of silver nanoparticles in many aspects of human life, it is extremely important to conduct more research aimed at revealing the mechanisms underlying their antibacterial properties. One of the strategies that allows for a more accurate investigation of the effect of antibacterial substances on cell morphology and metabolism is the application of the matrix-assisted laser desorption/ionization time-of-flight mass spectrometry (MALDI-TOF MS) technique for the analysis of bacterial lipids and proteins profiles.

The MALDI-TOF MS technique is a reliable tool for the analysis of microbial lipid profiles that allows for monitoring various metabolic processes occurring in the cell [45,46,84,85,86]. Lipids are the main component of cell membranes; thus, they correspond to the integrity of the cell as well as participating in the DNA modulation, acting as signaling molecules and providing a source of energy [46,84,85]. The main lipids present in the microbial cells are membrane phospholipids, especially glycerophosphocholines (PC), glycerophosphoethanolamines (PE), glycerophosphoglycerols (PG) and cardiolipins (CL) as well as their lyso-forms (e.g., LPGs, LPEs) [46,87]. Studies conducted by AlMasoud et al. [84] indicated that a direct analysis of lipids from intact bacteria cells by the MALDI technique was unsatisfactory. For this reason, in the present study, the chloroform/methanol (2:1) mixture was used to extract the intact lipid molecules from bacterial cells. The MALDI spectra for bacteria lipid extracts were recorded in the 190–2500 m/z range in both a positive and negative ion mode. In order to prove the veracity of the recorded signals, coming from the bacterial membrane lipids, the LIFT-TOF/TOF analysis was carried out. Moreover, the fragmentation path of selected lipids was described. Therefore, Figure 7A presents the MALDI spectra, as an example of lipid extracts, in positive ion mode of a protonated molecular ions [M + H]^+^ recorded at 429.260 m/z for *S. aureus* treated with ampicillin. The fragmentation pathway of 429.260 m/z presented in Figure 7B shows one dominant fragment ion at 215.818 m/z, which is related to the presence of a head group of phosphatidylglycerol (PG) [C_6_H_14_PO_6_ + H]^+^ [86]. The second, less intensive signal at 184.498 m/z is derived from the dodecan ion formed after PG(12:0/0:0) molecule fragmentation. Moreover, on the MALDI-TOF MS spectra for *S. aureus* treated with ampicillin, numerous signals derived from phospholipids of the same class (PG) but with different alkyl chain lengths were observed (Figure 7C). In addition, Figure 8A shows the MALDI spectra of another example of lipid extracts in the positive ion mode of a protonated molecular ion [M + H]^+^ recorded at 762.545 m/z for *P. aeruginosa* treated with silver ions; the probable fragmentation pathway is presented as well. According to Figure 8B, the most intensive signal on the fragmentation spectra can be observed at 184.368 m/z. This signal is assigned to the head-group of phosphatidylcholine [C_5_H_14_NPO_4_ + H]^+^ [88]. The other observed signals at 104.308 and 86.307 m/z corresponding to the choline [C_5_H_13_NO + H]^+^ and dehydrocholine [C_5_H_12_N]^+^, respectively [88]. In the case of *P. aeruginosa* treated with silver ions, as in the case of *S. aureus*, a series of signals from the same class of phospholipids (PC) differing only in the length of the alkyl chain was observed (Figure 8C).

In order to carry out a more thorough study, based on the changes occurring on the microorganisms’ profiles as a result of their exposure to stressors, a comprehensive analysis of signals changes on the recorded MALDI spectra was conducted. Therefore, the next crucial step of the study was to compare the lipids and protein profile of native bacteria cells with bacteria cells treated with AMP, TET, silver ions as well as non-functionalized and functionalized LCLB56_AgNPs. For this step, the bacterial cells suspended in MH (final density of 1 × 10^6^ CFU/mL) were treated with the given stress agent at a dose of 25% of the MIC value according to the Table 3. The dose was selected experimentally to ensure a sufficient number of the cells for MALDI analysis, after treatment with the selected concentration of silver (bio) nanoparticles. The differences in the signals observed on the lipids profile spectra were summarized in the Appendix A with the corresponding possible lipids assigned according to the LIPID MAPS^®^ Lipidomics Gateway database [44,89]. These tables only represent signals that came from the lipids generated by cells treated with antibacterial agents and that differed from native cells. It was observed that the bacterial cells growth in the presence of all stress agents leads to numerous changes in their membrane lipids composition. After cells being treated with antibacterial substances the disappearance of signals present on the spectra of native cells was noted. These signals come mainly from polyunsaturated membrane phospholipids. For example, the signal at 602.48 m/z that disappears after the incubation of *K. pneumoniae* with all silver-based antibacterial agents comes from unsaturated glycerophosphoglycerol ([PG(66:4(OH))-H]^2−^).

In the case of *K. pneumoniae* (Appendix A), the signal recorded at 498.644 m/z, which disappears after treatment with almost all tested agents, except the LCLB56_AgNPs and LCLB56_AgNPs/AMP, can be related with the degradation of another unsaturated cardiolipin ([CL(76:113)-3H]^3−^). Another example can be the disappearance of signal at 353.59 m/z after the incubation of *K. pneumoniae* with silver ions related to the degradation of unsaturated glycerophosphoethanolamine ([PE(60:10)-3H]^3−^). The disappearance of these signals may be connected to the peroxidation of unsaturated double bonds of fatty acids, which may result in the loss of cell membrane integrity. This effect was previously observed by Calvano et al. [44] as a result of exposure to *E. coli* cells to copper salts. However, in addition to the disappearance of signals on the MALDI spectrum due to the incubation of cells with stress factors, in our studies, we also observed the appearance of many new signals. The new recorded signals come from three major types of lipids. The first type of lipids are those with a hydroxyl group, e.g., after *E. coli* (Appendix A) incubation with silver-based substances, the appearance of signals at 328.895, 330.889, 346.685 and 598.308 m/z correspond to [PE(53:6(OH))-3H]^3−^, [PE(54:10(OH))-3H]^3−^, [PE(32:6(OH))-2H]^2−^ and [PE(20:0(OH)) + OAc]^−^, respectively. The second group of appearing lipids represents ether lipids, e.g., after the application all of tested compounds against *S. aureus* (Appendix A), the new signal in comparison to native bacteria cells at 303.247, 416.359, 430.386, 453.477, 460.425 and 472.473 related with [PG(O-47:3)-3H]^3−^, [DG(O-49:3) + 2Na]^2+^, [TG(O-53:2) + 2H]^2+^, [DG(O-26:0) + H-H_2_O]^+^, [TG(O-58:7) + 2H]^2+^ and [TG(O-59:2) + 2H]^2+^, respectively, was noted. The ether lipids are a specific class of glycerophospholipids in which the alkyl chain is attached by an ether bond to the *sn-1* position [90,91]. The presence of this type of lipid significantly affects the morphology of the cell membrane. The large amount of these compounds enables tighter phospholipid packing in the membrane, which results in its reduced fluidity and increased rigidity. Moreover, it is believed that ether lipids can function as endogenous antioxidants [90,91]. Therefore, the increased production of these lipids constitutes a cell defense mechanism, which on the one hand reduces the permeability of cell membrane, and on the other protects cells against oxidative stress caused by silver ions and silver (bio) nanoparticles. The last observed group consist of lysophospholipids, e.g., after the incubation of *K. pneumoniae* with both non-functionalized and functionalized silver (bio) nanoparticles, the new signal at 242.115, 503.347, 566.378, 580.343 and 582.348 m/z associated with [LPA(22:6) + 2H]^2+^, [LPA(24:1) + H-H_2_O]^+^, [LPS(21:0)-H]^−^, [LPE(26:6)-H]^−^, [LPE(26:5)-H]^−^ and [LPS(24:5)-H]^−^, respectively, was observed. Lysolipids are nonbilayer-forming lipids, which can only be found in normal bacterial cell membranes in a trace amount. Its accumulation in the cell membrane can disrupt its structure through the increase in its permeability and the induction of the curvature. This effect of lysolipids is related with their detergent-like physical properties [92,93]. Therefore, the appearance of a large amount of this class of lipids after the incubation of bacteria with antibacterial agents may indicate that the mechanism of the antimicrobial action of tested compounds is based on the destruction of the integrity of biological cells barriers. This result is in agreement with the results obtained by the flow cytometry, which showed the formation of fragments of disrupted bacteria cells due to incubation with functionalized and non-functionalized silver (bio) nanoparticles.

Moreover, the investigation of bacteria lipids’ profile was previously described as a method for testing microbial resistance. Kuyukina et al. [94] found that the lipids’ composition of rhodococci could provide useful information to determine their sensitivity to antibiotics. They noted that the increase in the amount of total lipids present in the cell as well as the appearance of larger amounts of cardiolipin and phosphatidylglycerol is correlated with the enhanced antibiotic resistance of these bacteria. A similar phenomenon can be observed in our study. The number of new signals after the incubation of cells with antibacterial agents is definitely higher as compared to the disappearing signals, which can indicate the increased lipid production. In addition, many new signals appearing after cell incubation with antibacterial agents can be assigned to cardiolipins (e.g., signal at 427.936 m/z observed for *E. coli*, *S. epidermidis*, and *P. aeruginosa*) and phosphatidylglycerols (e.g., signals at 429.994 m/z observed for *S. aureus*, *S. epidermidis*, *K. pneumoniae*, and *P. aeruginosa*; Figure 7). Although the number of these changes is not very high, their presence may indicate the start of the activation of drug resistance mechanisms by the tested cells.

The MALDI technique is also widely used in the study of the protein profiles of microorganisms for their identification as well as the tracking of their metabolic processes [8,44,47,86]. Many studies confirmed that this technique allowed for the monitoring of changes in bacterial cells arising as a consequence of exposure to various stressors. Hasan et al. [43] used this technique to study the reaction of *E. coli* on heat stress. In turn, Rafińska et al. [8] used the MALDI to study the changes in *B. subtilis* cells as a result of exposure to silver ions, silver nanoparticles, and tetracycline. Calvano et al. [44] studied the changes in the *E. coli* protein profile under the influence of copper ions and nanoparticles. 

Therefore, in the present study, the differences in the signals observed in the recorded spectra with the corresponding possible proteins assigned according to the UniProt database are summarized and presented in Appendix A. As can be observed, the incubation of all studied bacteria strains with tested antibacterial agents leads to numerous changes in the protein expression. In the case of *E. coli* (Appendix A), the cells’ incubation with all tested compounds results in the lack of the expression of the RNA helicase protein (2341 m/z)—an enzyme that participates in almost all aspects of the RNA metabolism [95]. Furthermore, after cells’ incubation with silver-based substances, the appearance of a new signal at 3411 m/z, probably deriving from the DNA gyrase inhibitor, was noted. The presence of this protein inhibits the action of the DNA gyrase—an essential enzyme in the process of the DNA replication, transcription, and chromosome segregation [96]. Similarly, after the application of LCLB_AgNPs, the signal at 4468 m/z appears; it probably corresponds to ribonucleotide reductase transcriptional regulator, which causes a negative regulation of transcription [97]. Moreover, after cells’ treatment with tetracycline-functionalized silver (bio) nanoparticles, the new signal at 4578 m/z can be observed. This signal can be related with the lysis protein and the presence of this protein may result in the degradation of the cell membrane of a bacterial cell. A protein with a similar function— holin protein (7964 m/z)—also appears after using tetracycline [98]. The loss of the cell membrane integrity after using tetracycline-functionalized (bio) nanoparticles, also confirmed by the flow cytometry (the appearance of disrupted cells fragments), can therefore be the result of the synergistic action of nanocomposites and tetracycline. 

In the case of *S. epidermidis* protein profiles, many changes in the spectrum after its incubation with stress agents were also observed (Appendix A). However, most of the recorded signals probably originate from newly discovered yet uncharacterized proteins. Only a few signals could be attributed to specific proteins. An example was the emergence of new signals at 4305, 5946 and 6516 m/z after the application of all tested compounds related to the presence of 50 S ribosomal proteins, which are a structural constituent of ribosome [99]. In addition, after exposure to all studied stress agents, the appearance of signal at 10,429 m/z associated with the expression of formate dehydrogenase was noted. This protein possesses the oxidoreductase activity and its appearance may be related to the cell’s response to oxidative stress [100].

Most changes in the protein molecular profile of all tested strains were recorded for *S. aureus* (Appendix A). The most important changes in the protein profile of this bacteria after its exposure to tested substances include the lack of expression of the GNAT family N-acetyltransferase protein (3183 m/z) responsible for the glucose metabolic process, which can disrupt cell function [101]. Another important change in the protein profile of *S. aureus* was the loss of signal at 4764 m/z after cell incubation with all of tested compounds associated with the absence of amino acid permease, which is responsible for the intermembrane transport of amino acids [102]. Therefore, the lack of the expression of this protein may be associated with the activation of cell defense mechanisms that prevent the transport of toxic substances into its interior. Moreover, as with *S. epidermidis*, the appearance of signals from 50 S ribosomal proteins (4291, 4871, 59329734, 9758, and 9797 m/z) after the application of all tested antibacterial agents with the exception of ampicillin was observed. A new signal at 7021 m/z after the cell exposure to LCLB56_AgNPs can be observed. It can be assigned to thiol peroxidase, Bcp-type responsible for peroxidase activity, and therefore it responds to oxidative stress [103]. On the spectra of cells treated with non- and tetracycline-functionalized LCLB56_AgNPs, the appearance of a signal at 7713 m/s, coming probably from prolipoprotein diacylglyceryl transferase, can be also noted. The presence of this protein is caused by the lipoprotein biosynthetic process and may be responsible for some of the changes recorded at the MALDI spectra recorded for lipids profile [104]. In addition, after the cell incubation with tetracycline-functionalized LCLB56_AgNPs, the appearance of the signal related with the RecA protein (9911 m/z) responsible for the DNA repair process was noted [105].

Moreover, numerous changes in the protein profile of *P. aeruginosa* cells were observed (Appendix A). The most important include the signal loss at 5273 and 7328 m/z after the application of functionalized and non-functionalized LCLB56_AgNPs derived from monovalent cation/H+ antiporter subunit F and copper chaperone [106]. Both of these proteins are responsible for metal ion transport and their lack may be caused by a defensive reaction preventing the transport of silver ions into the cell. Moreover, after the application of non- and tetracycline-functionalized LCLB56_AgNPs, the disappearance of signal at 7025 m/z was noted. This signal can be attributed to cytochrome c protein, and the lack of its expression can lead to a loss of the capacity for the anaerobic respiration of the cell [107].

In the case of *K. pneumoniae* (Appendix A), the disappearance of signal at 3580 m/z after the cell incubation with LCLB56_AgNPs functionalized with both antibiotic can be seen. It can be probably connected with the cessation of the AP endonuclease production. This enzyme is responsible for repairing damaged or mismatched nucleotides in the DNA and its absence may be the result of the cell damage [108]. After the application of silver (bio) nanocomposites functionalized with ampicillin and tetracycline the appearance of signals from heat- (3912 m/z) and cold-shock (4143 m/z) protein, respectively, was noted. Moreover, the cell incubation with LCLB56_AgNPs leads to the expression of entericidin B (4810 m/z) and klebicin B immunity (9618 m/z). The entericidin B expression is a response of the cell to the exposure to toxic substances, and the klebicin B immunity is a protein responsible for toxic substances binding [109]. Therefore, the appearance of these proteins may indicate the activation of cell defense mechanisms. In addition, the exposure to the same substance causes the appearance of the protein responsible for repairing the damaged DNA (DNA repair protein RadA, 6137 m/z) [110].

On the basis of the detected proteins generated by the UniProt database, it was noticed that some of the identified proteins were common for different bacteria strains (S11) For example, the presence of 50 S ribosomal protein L36 and L33 for both *S. aureus* and *S. epidermidis* bacteria treated with LCLB56_AgNPs was noted. In addition, the proteins of the transposase family responsible for the DNA integration appeared after the application of LCLB56_AgNPs/AMP in cells of *S. aureus, P. aeruginosa and K. pneumoniae*. In turn, incubation with LCLB56_AgNPs/TET resulted in the DNA-binding protein expression in the case of *E. coli*, *S. aureus* and *P. aeruginosa*. None of the differentiating signals could be identified as virulence factor or proteins associated with cell membrane metabolism. However, to obtain an accurate characterization of unidentified signals, their further analysis using 2D gel electrophoresis or the nano-LC method is necessary since used MALDI TOF MS protocol is dedicated to sufficient bacterial molecular fingerprint comparison, the detection of differentiating signals and, if possible, giving the most probable name of the molecule responsible for signal, which still needs to be confirmed by other methods.

In order to compare the effect of the incubation of cells with a given antibacterial agent on their protein and lipid profile, a multivariate statistical analysis of samples was performed based on the differentiating signals observed on the MALDI spectrum (Appendix A). Figure 9 presents the impact of a given antiseptic agent on the protein and lipid profiles of all tested bacteria based on their grouping on the PC-plane. The performed analysis revealed a great impact of the antiseptic agent addition on the profiles of investigated biomolecules which significantly varied between bacterial strains. Considering proteins, the biggest difference was observed for *S. aureus* and *P. aeruginosa* strains (Figure 9A). In the case of *S. aureus*, cells treated with LCLB56_AgNPs, LCLB56_AgNPs/AMP, and LCLB56_AgNPs/TET considerably differed in specific protein signals from the other experimental variants wherein the influence of silver (bio) nanoparticles was opposite to that observed for the antibiotic. Regarding the changes in the protein profiles of *P. aeruginosa*, the presence of two main groups could be observed. The first one consist of cells treated with LCLB56_AgNPs (non-functionalized) and LCLB56_AgNPs/AMP (functionalized), while the second one consist of native cells and cells treated with antibiotics, silver ions and LCLB56_AgNPs/TET (functionalized). Changes in the protein profiles of the rest strains—*S. epidermidis*, *E. coli*, and *K. pneumoniae*—were significantly smaller compared to the first two strains; nevertheless, a different impact of the silver (bio) nanoparticles as compared to the antibiotics alone can be observed. In particular, it was evident in the case *S. epidermidis* and *E. coli,* where a similar division between cells treated with all silver (bio) nanoparticles forms, and the rest variants were revealed. Furthermore, for *S. epidermidis,* the protein profiles of the bacteria treated with the AMP were most similar to native cells, while TET-treated cells constituted a separate group. In addition, cells of this bacteria incubated with all silver-based substance protein profiles showed considerable similarity. The *E. coli* and *K. pneumoniae* protein profiles of cells after its incubation with LCLB56_AgNPs/AMP, and LCLB56_AgNPs/TET showed great similarity, which would suggest a similar mechanism of action of these factors in relation to these bacteria. Moreover, the *E. coli*, *S. aureus* and *K. pneumoniae* protein profiles after cells treated with both antibiotics show the highest similarity. Regarding lipids profiles, the performed analysis revealed big differences in the changes of their composition among investigated bacterial strains (Figure 9B). Unlike the changes in protein profiles, the largest differences were observed for *K. pneumoniae* and *S. epidermidis* cells. For the first strain, a clear division into two clusters could be observed—cells treated with functionalized and non-functionalized LCLB56_AgNPs as well as those exposed to antibiotics and silver ions. The lipid profiles of both groups considerably differed from the ones of native cells. In the case of *S. epidermidis*, a considerably lower impact of ampicillin and tetracycline addition on the changes in the lipids profiles was revealed as compared to all the silver forms, including Ag^+^. A similar effect was noted for *E. coli* cells; however, observed differences were significantly lower. As in the case of the investigation of protein profiles changes, *S. aureus* and *P. aeruginosa* demonstrated a similar response to the antibacterial agent addition on the composition of lipids. Although both strains varied in lipids composition, they revealed the same division on the native cells and those treated with antibacterial agents without revealing large differences between individual treatment options. In general, both functionalized and non-functionalized silver (bio) nanoparticles exerted the greatest influence on changes in the protein and lipid profiles of tested microorganisms whose significance was strongly strain-dependent. The analysis revealed a bigger variance in the lipid profiles compared to the protein profiles; however, regardless of the type of tested biomolecules, *S. aureus* and *P. aeruginosa* revealed a similar kind of response to the antibacterial agent addition. In case of registred lipid profiles, the application of all of the tested antibacterial agents led to intense differences compared to untreated cells, while, in case of protein profiles, similarity has been observed in the changes induced by the respective bacterial strains incubated with LCLB56_AgNPs, LCLB56_AgNPs/AMP and TET. It was noticed that the bacterial cells treated with non-functionalized (bio) nanoparticles and functionalized with ampicillin form a separate group, whereas the cells treated with TET generated a group separated from the others tested agents.

## 4. Conclusions

Our results reveal that silver (bio) nanoparticle functionalization is a complex process of different nature and effectiveness depending on the used antibiotic. For ampicillin, a thick layer of antibiotic was observed on the surface of the (bio) nanoparticles, whereas, for tetracycline, intramolecular diffusion was noted. The tested nanocomposites before and after functionalization showed dispersion stability at pH above 6. The minimum inhibitory concentration determination and flow cytometry analysis allowed for studying the antibacterial activity of the tested complexes against a range of Gram-negative and Gram-positive pathogenic bacteria strains. The increase in the biological activity was observed only in the case of *S. epidermidis* treated with LCLB56_AgNPs/AMP. In other cases, the functionalization did not improve the antibacterial properties of (bio) nanoparticles, and, in some instances, it even caused a slight decrease. The MALDI-TOF MS technique turned out to be a useful tool to further explain the mechanisms underlying the antibacterial effects of the test substances. This technique allowed for the identification of changes in the lipid and protein profile of microorganisms exposed to stressors. The observed changes in the spectra indicate both the activation of cell defense mechanisms and their damage. One of the main possible defense mechanisms was the reduction in cell membrane permeability both through the synthesis of ether lipids and the discontinuation of the production of proteins responsible for intermembrane transport. However, a long-term exposure to antibacterial agents resulted in a loss of the cell membrane integrity and, consequently, the cell death, as evidenced by the results obtained by the flow cytometry as well as the increase in lysolipids production and expression of proteins responsible for cell lysis. Moreover, the PCA analysis of the obtained protein and lipid bacteria profiles showed significant differences at the molecular level of the same microbial species treated with non-functionalized and functionalized (bio) nanoparticles. Therefore, it can be concluded that, although the functionalization of silver nanocomposites with antibiotics did not significantly affect the obtained MIC value, it had a significant impact on the mechanism of (bio) nanoparticles’ action on bacterial cells on a molecular level.

## Figures and Tables

**Figure 1 materials-13-02403-f001:**
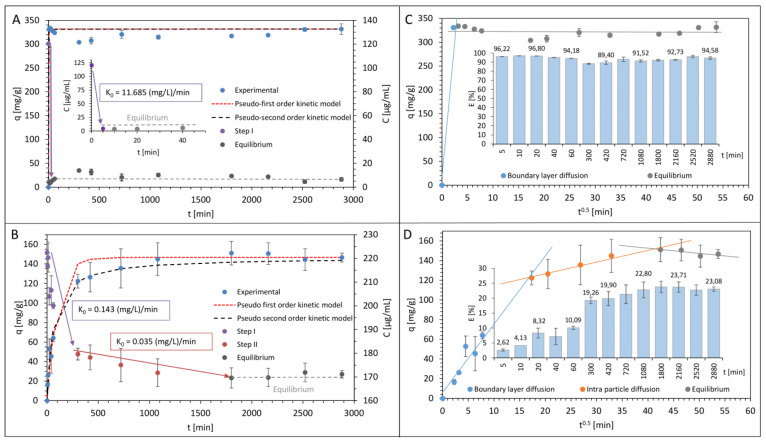
Experimental kinetics data and fitted pseudo-first- and -second-order kinetic models as well as the kinetic steps of the sorption and values of the rate constants determined using a zero-order kinetic model for ampicillin (**A**) and tetracycline (**B**). The plot of the intra-particle diffusion model and sorption effectiveness for ampicillin (**C**) and tetracycline (**D**). k_0_ is the rate constant of the zero-order kinetic model; q is the amount of the antibiotic sorbed in each time period (t); C is the antibiotic concentration in supernatant at each time period (t); E% is the sorption effectiveness.

**Figure 2 materials-13-02403-f002:**
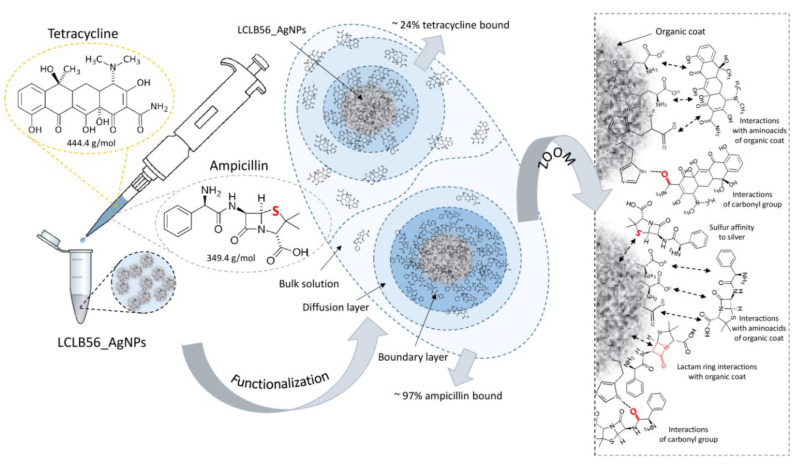
Proposed mechanism of LCLB56 silver (bio) nanoparticles with ampicillin and tetracycline.

**Figure 3 materials-13-02403-f003:**
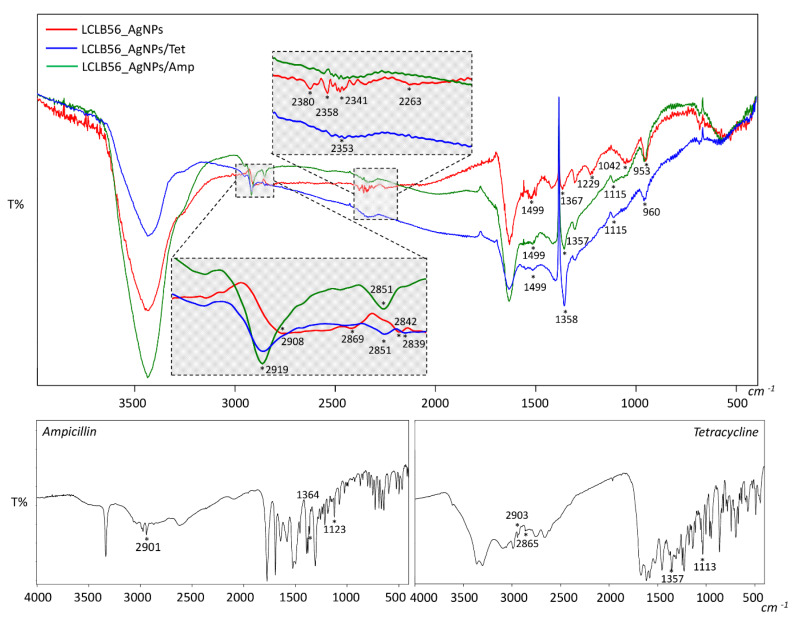
Fourier transform infrared spectroscopy (FT-IR) spectra of LCLB56_AgNPs functionalized and non-functionalized with antibiotics as well as the FT-IR spectra of ampicillin and tetracycline.

**Figure 4 materials-13-02403-f004:**
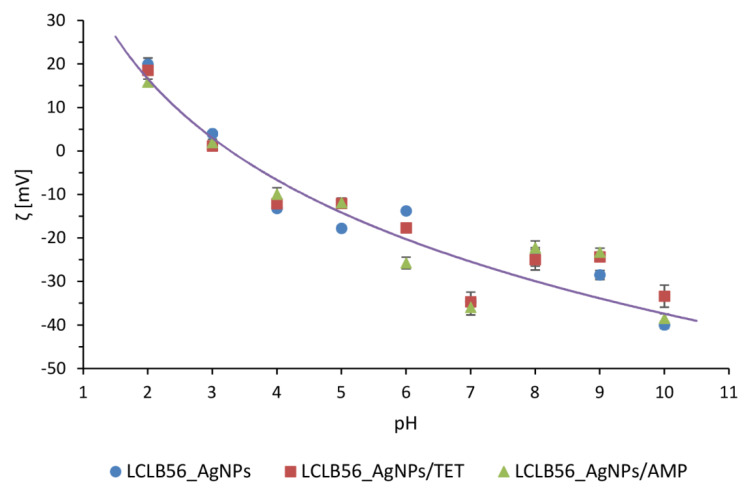
Comparison of the average zeta potential value of LCLB56_AgNPs non-functionalized and functionalized with antibiotics at different pH conditions.

**Figure 5 materials-13-02403-f005:**
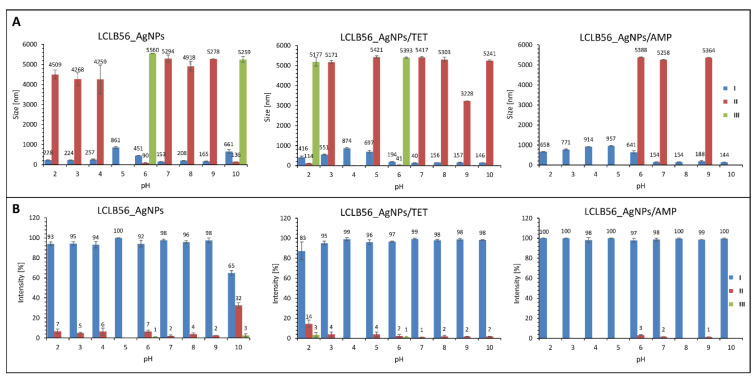
Hydrodynamic size distribution (**A**) and intensity (**B**) of the recorded populations (I, II III), on pH, of LCLB56_AgNPs, LCLB56_AgNPs/tetracycline (TET) and LCLB56_AgNPs/ampicillin (AMP).

**Figure 6 materials-13-02403-f006:**
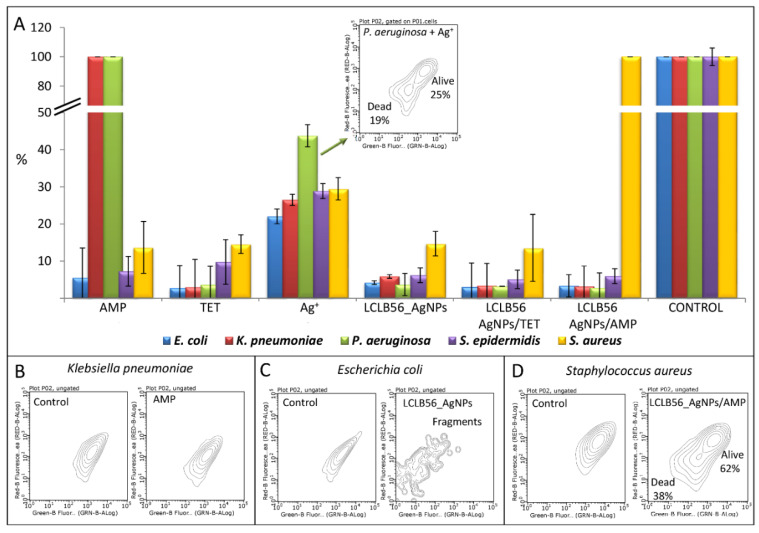
Change plot of the percentage of the total number of cells that grew after the 24-h incubation of bacteria with AMP, TET, silver ions and LCLB56_AgNPs functionalized and non-functionalized with antibiotics as well as without any stress agent. Control (**A**) examples of cells’ fluorescence after its incubation with and without a given stress agent (**B**–**D**).

**Figure 7 materials-13-02403-f007:**
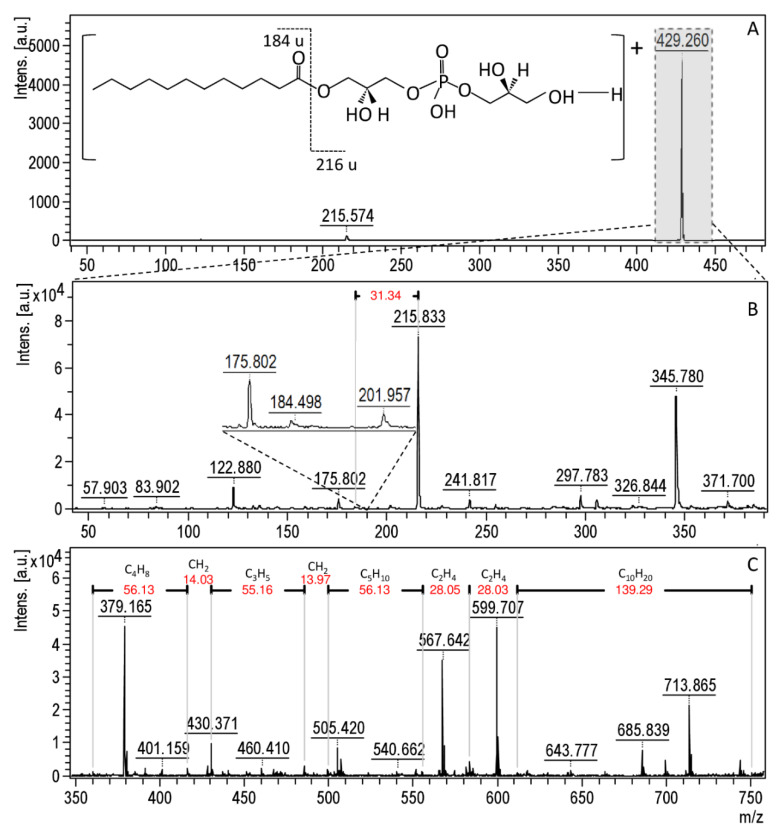
Fragmentation pathway of [PG(12:0/0:0) + H]^+^ m/z 429 and mass spectrometry (MS)/MS spectrum of the parent ion (**A**); MS/MS spectrum of fragmentation of signal at 429 m/z (**B**); MALDI-TOF MS spectra obtained for *S. aureus* after incubation with ampicillin (**C**).

**Figure 8 materials-13-02403-f008:**
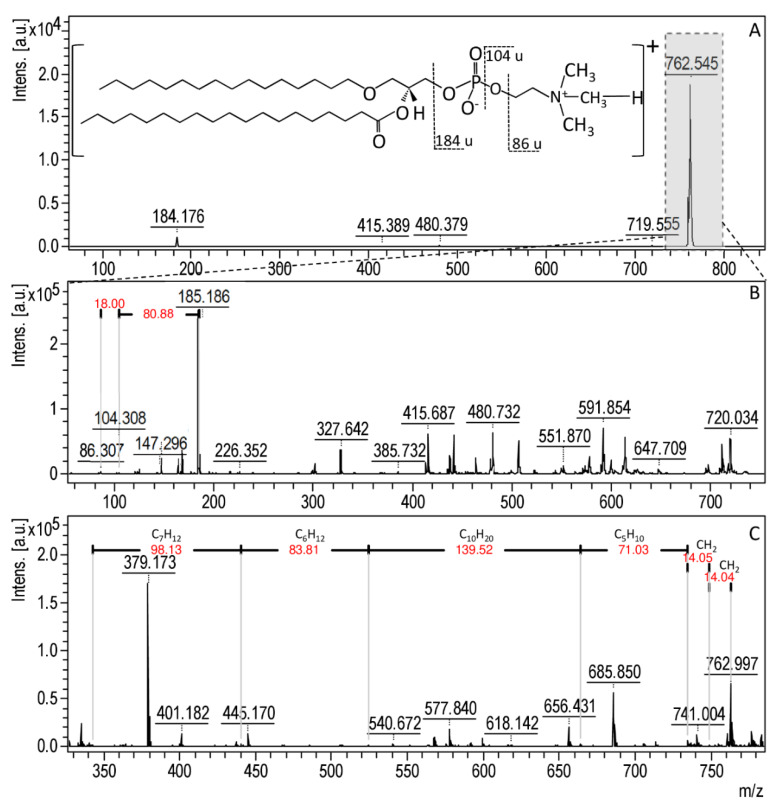
Fragmentation pathway of [PC(O-16:0/19:0) + H]^+^ m/z 762 and MS/MS spectrum of parent ion (**A**); MS/MS spectrum of fragmentation of signal at 762 m/z (**B**); MALDI-TOF MS spectra obtained for *P. aeruginosa* after incubation with silver ions (**C**).

**Figure 9 materials-13-02403-f009:**
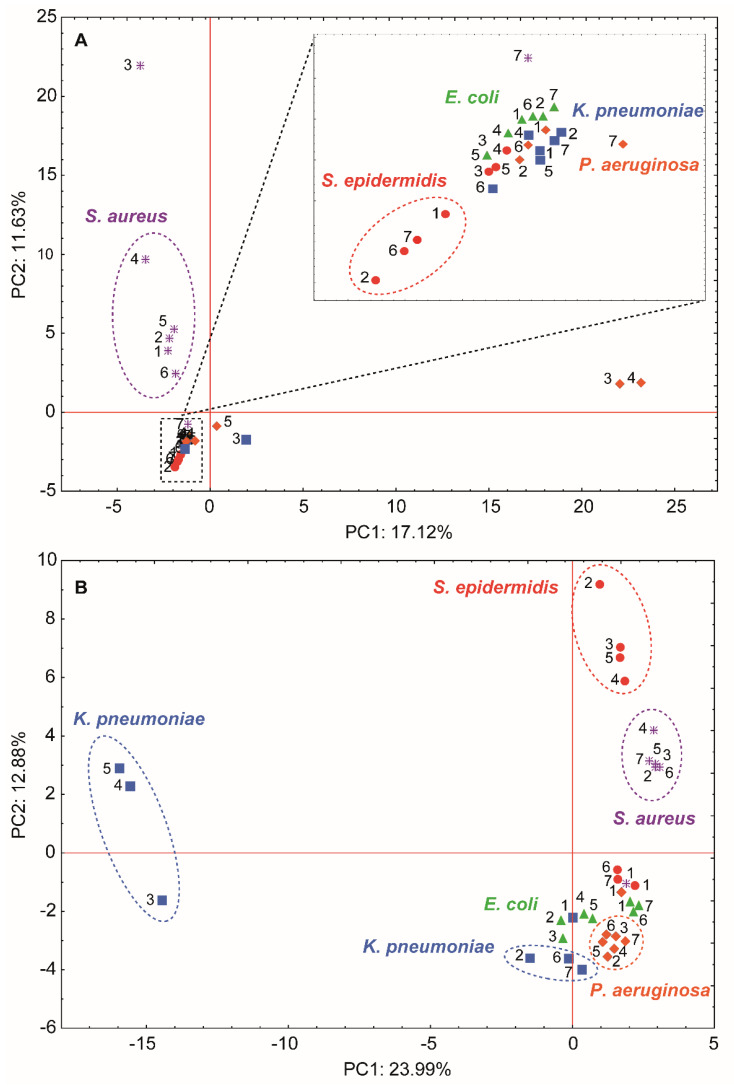
Impact of the silver ions (**2**), LCLB56_AgNPs (**3**), LCLB56_AgNPs/AMP (**4**) LCLB56_AgNPs/TET (**5**) AMP (**6**) TET (**7**) on the changes in signals observed on MALDI-TOF MS spectra recorded for proteins (**A**) and lipids (**B**) compared to the untreated bacterial cells (**1**) based on the grouping on the Factors-plane in PCA analysis.

**Table 1 materials-13-02403-t001:** Kinetic model parameters for the antibiotics’ sorption by LCLB56_AgNPs.

Zero-Order Kinetics Model		Pseudo-First-Order Kinetics Model		Pseudo-Second-Order Kinetics Model		Intra-Particle Diffusion Model	
**Ampicillin**							
**First Step** **k_0_ (mg L^−1^ min^−1^)**	11.685	q_e_ (mg g^−1^) k_1_ (min^−1^) A_approx._ (%)	331.486 0.644 2.344	q_e_ (mg g^−1^) k_2_ (min^−1^) A_approx._ (%)	331.484 0.028 2.339	A (mg g^−1^) K_ip_ (mg g^−1^ min^−0.5^)	322.800 0.034
**Tetracycline**							
**First Step** **k_0_ (mg L^−1^ min^−1^)**	0.143	q_e_ (mg g^−1^) k_1_ (min^−1^) A_approx._ (%)	146.714 0.010 19.968	q_e_ (mg g^−1^) k_2_ (min^−1^) A_approx._ (%)	143.654 1.111 × 10^−4^ 8.571	A [mg g^−1^] K_ip_ (mg g^−1^ min^−0.5^)	102.950 1.186
**Second Step** **k_0_ (mg L^−1^ min^−1^)**	0.035						

k_0_ is the rate constant of the zero-order kinetic model; k_1_ is the rate constant of the pseudo-first-order kinetic model; k_2_ is the rate constant of the pseudo-second-order kinetic model; q_e_ is the amount of the antibiotic sorbed in equilibrium; A_approx._ is the relative approximate error; A is a constant related with the thickness of the boundary diffusion layer or the adsorption of the external surface; K_ip_ is the diffusion rate constant.

**Table 2 materials-13-02403-t002:** Values of the distribution coefficient and the change Gibbs free energy of the antibiotics sorption.

q_e_ (mg/kg)	Ce (mg/L)	K_d_	T (K)	ΔG_0_ (kJ mol^−1^)
**Ampicillin**				
**331486.419**	6.521	50835.086	295	−26.578
**Tetracycline**				
**146713.500**	171.170	857.123	295	−16.564

q_e_ is the amount of antibiotic sorbed in equilibrium; C_e_ is the equilibrium concentration of antibiotic in solution; K_d_ is the distribution coefficient; T is temperature; ΔG_0_ is the change of the Gibbs free energy.

**Table 3 materials-13-02403-t003:** Minimum inhibitory concentration (MIC) of tested antimicrobial agents.

	Minimum Inhibitory Concentration [µg/mL]					
	AMP	TET	Ag^+^	LCLB56 AgNPs	LCLB56 AgNPs/AMP	LCLB56 AgNPs/TET
*E. coli*	6.25	3.125	3.125	6.25	12.5	12.5
*S. aureus*	12.5	1.56	12.5	12.5	50	12.5
*S. epidermidis*	12.5	6.25	6.25	12.5	6.25	12.5
*K. pneumoniae*	50	0.78	6.25	12.5	25	12.5
*P. aeruginosa*	–	0.78	1.56	12.5	12.5	12.5

AMP is ampicillin; TET is tetracycline; Ag^+^ is silver ions; LCLB56AgNPs is silver (bio) nanoparticles synthesized by L. lactis; LCLB56AgNPs/AMP is silver nanocomposites functionalized with ampicillin; LCLB56AgNPs/TET is silver nanocomposites functionalized with tetracycline.

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
