# Peer review of "The Influence of Different Forms of Silver on Selected Pathogenic Bacteria"

_materials, 2020, doi:10.3390/ma13102403_

Round 1

Reviewer 1 Report

In general, the manuscript is adequately prepared but some minor revisions in the language used need attention:

line 17: change "For this purpose the" to " For this purpose, the "

line 18: change " formulations was characterized " to " formulations were characterized "

line 177: change “The spectra was recorded” to “The spectra were recorded”

line 236: change “The spectra was recorded” to “The spectra were recorded”

line 458: change “However,the” to “ However, the”

line 699: change “Similarly after” to “Similarly, after”

lines 723 the word “permease” seem to be wrong edited

line 730: change “it respond” to “it responds”

line 779: change “the biggest differences was observed” to “the biggest difference was observed”

Also in the literature there are relevant papers that refer to the bio synthesis of silver nanoparticles that should be cited:

Mat. Sci. Eng. C 101, 120-137, 2019

Journal of Nanomaterials, vol. 2017, Article ID 4214017, 12 pages, 2017

Author Response

Response to Reviewer 1 Comments

Comments of the Reviewer [materials-811885]

We would like to thank the Editor and Reviewer for careful reading, and constructive suggestions for our manuscript that will help us to improve our work. According to this, we have comprehensively revised our manuscript. Hoping that we have addressed all the questions mentioned by the Reviewer, below we included point-to-point the response to each comments.

In manuscript file all of the changes have been provided in red color.

REVIEWER COMMENTS:

Point 1: In general, the manuscript is adequately prepared but some minor revisions in the language used need attention:

line 17: change "For this purpose the" to " For this purpose, the "

line 18: change " formulations was characterized " to " formulations were characterized "

line 177: change “The spectra was recorded” to “The spectra were recorded”

line 236: change “The spectra was recorded” to “The spectra were recorded”

line 458: change “However,the” to “ However, the”

line 699: change “Similarly after” to “Similarly, after”

lines 723 the word “permease” seem to be wrong edited

line 730: change “it respond” to “it responds”

line 779: change “the biggest differences was observed” to “the biggest difference was observed”

Response 1: All comments mentioned by the Reviewer have been taken into account and the suggested forms were provided in the manuscript. Concerning the line 723 regarding the word “permease”, the author has improved the sentence in order to avoid misunderstanding.

Point 2: Also in the literature there are relevant papers that refer to the bio synthesis of silver nanoparticles that should be cited:

Mat. Sci. Eng. C 101, 120-137, 2019

Journal of Nanomaterials, vol. 2017, Article ID 4214017, 12 pages, 2017

Response 2: We would like to thank for pointing out interesting works in the field of biological synthesis of silver nanoparticles. Taking in consideration the Reviewer suggestion the mentioned papers was cited in the manuscript.

Reviewer 2 Report

The article "The influence of different form of silver on selected pathogenic bacteria" makes a somewhat mixed impression. The methods are sound, the experimental quality is high, and the amount of work is huge indeed, however the overall impact is not big.

The essential conclusion from the work is that the activity of silver nanoparticles alone, antibiotics alone or their combinations is nearly same in terms of MIC. The after all article shows that there is no synergetic effect between the active components. In the methabolic pathways there some differences in bacterial compositions, however mass-spectrometry alone is not a method reliable enough to draw firm conclusions about protein's structures of the treated bacteria. It also can be somewhat understood that different antimicrobial agents would affect different bacterial organella.

The article calls for some questions regarding the AgNP and their stability. On line 407 authors claim that the measured size (hydrodynamic) of NP is larger because of solvatation. However, in their case the size grows from 5-50 nm as measured by TEM to 100-1000 nm as determined by DLS. Indeed, this is too much or solvataion and should be attributed to aggregation of the particles during evaporation in DLS spray.

Another questionable point is sorption of antibiotics and chemical composition of the NP shell. In lines 277-289 authors discuss "molecular mass and surface area" of ampicillin and tetracycline. Leaving aside their reference "according to PubChem" (line 284), the surface area will differ depending on the orientation of the molecule. Taking into account the protein shell of unknown thickness and composition surrounding the NP, the whole discussion does not look convincing.  

In conclusion, I would recommend to publish the article after revision. It might be a good example of a well proved negative result, however should be rewritten to focus on essential points.

Author Response

Response to Reviewer 2 Comments

Comments of the Reviewer [materials-811885]

We would like to thank the Editor and Reviewer for careful reading, and constructive suggestions for our manuscript that will help us to improve our work. According to this, we have comprehensively revised our manuscript. Hoping that we have addressed all the questions mentioned by the Reviewer, below we included point-to-point the response to each comments.

In manuscript file all of the changes have been provided in red color.

REVIEWER COMMENTS:

The article "The influence of different form of silver on selected pathogenic bacteria" makes a somewhat mixed impression. The methods are sound, the experimental quality is high, and the amount of work is huge indeed, however the overall impact is not big.

Point 1: The essential conclusion from the work is that the activity of silver nanoparticles alone, antibiotics alone or their combinations is nearly same in terms of MIC. The after all article shows that there is no synergetic effect between the active components. In the methabolic pathways there some differences in bacterial compositions, however mass-spectrometry alone is not a method reliable enough to draw firm conclusions about protein's structures of the treated bacteria. It also can be somewhat understood that different antimicrobial agents would affect different bacterial organella.

Response 1:. Yes, the authors are agree that the mass-spectrometry alone is not a method reliable enough to draw firm conclusions about protein's structures, their sequencing. Such studies can be performed using, for example, the positive (or negative) reflectron mod in combination with e.g. 1D or 2D -gel electrophoresis (bottom up approach) or Shotgun analysis and this will be the subject of our research in the future. However, once the MALDI technique using soft ionization enables to register specific molecular profile of microorganism such as bacteria (Calvano et al.; J. Mass Spectrom. 2016, 828–840), in present study, the respective technique was applied to investigate cellular changes at the molecular fingerprint (mostly protein) level only, not individual protein’s structure. Therefore, this technique was employed as a complementary method to investigate the changes in protein and lipid profile of bacteria treated with different stress agents.

Point 2: The article calls for some questions regarding the AgNP and their stability. On line 407 authors claim that the measured size (hydrodynamic) of NP is larger because of solvatation. However, in their case the size grows from 5-50 nm as measured by TEM to 100-1000 nm as determined by DLS. Indeed, this is too much or solvataion and should be attributed to aggregation of the particles during evaporation in DLS spray.

Response 2: The DLS technique in the nano-sizer system allows the measurement of hydrodynamic size directly in a solution in the folded capillary U-cells (https://www.malvernstore.com/en-gb/categories/consumables/cells,-cuvettes-and-cups/DTS1070). Indeed, the aggregation effects in the test solution may also be involved in the process. However, the size value is closely correlated with the pH, type of solvent and zeta potential value. Moreover, a component affecting on the DLS size is brunched solvated organic coat, naturally presented into/onto silver nanoparticles core. According to the Reviewer’s remark, this information was included in the work.

Point 3: Another questionable point is sorption of antibiotics and chemical composition of the NP shell. In lines 277-289 authors discuss "molecular mass and surface area" of ampicillin and tetracycline. Leaving aside their reference "according to PubChem" (line 284), the surface area will differ depending on the orientation of the molecule. Taking into account the protein shell of unknown thickness and composition surrounding the NP, the whole discussion does not look convincing. 

Response 3: We would like to thank the Reviewer for this valuable remark. We agree with the Reviewer that discussion about the surface area of antibiotics in such a complex system was quite unfortunate. Taking this into account, the authors decided to base the discussion only on differences in the mass, structure and chemical composition of the used antibiotics. Appropriate changes have been introduced to the manuscript.

Point 4: In conclusion, I would recommend to publish the article after revision. It might be a good example of a well proved negative result, however should be rewritten to focus on essential points.

Response 4: According to the Reviewer remark, the authors have performed some changes in the manuscript that will better highlight the essential points of the work.

Reviewer 3 Report

The manuscript describes the development, and characterization both physico-chemical and biological of silver nanoparticles functionalized with two different antibiotics. The strategy is not new but results are puzzling in the literature. In this study, the authors performed an in-depth study of their systems. The methods and the processes used are fine and well performed. The results are not spectacular but anyway interesting in the field. After minor corrections the manuscript can be accepted in Materials for me. However, the manuscript has to be read-proof by a native speaker since there is a lot of grammar mistakes. Below are my main comments and minor corrections.

Main comments:

Figure 5: The choice of representation for the graphs are not appropriate. Please choose something like histograms. In the current version, we cannot visualize all the data. A clear figure is crucial since this part is tough to understand from the text.

Table 3: Although it seems clear that the concentrations determined for the MIC values in µg/mL correspond to the concentration in Ag or in the antibiotic, but in the case of functionnalized AgNPs, this is unclear. This point has to be clarified and explained how it is calculated since it is crucial to draw proper conclusions concerning these MIC comparisons.

The same applies for the data presented in Figure 6, in the current form, it is very difficult to conclude about the functionalized AgNPs. The experiments presented in Figure 6 are also not properly designed. Indeed, non-functionalized AgNPs are already highly effective in killing bacteria at 25 µg/mL. It is therefore impossible to see an additive effect in the case of the functionalized AgNPs. This experiment has to be redone at lower concentrations.

The concentration(s) used for the mass spectrometry experiments is not specified in the main text. This is missing as well as the rationale for the choice of this dose.

Minor corrections.

In the abstract, line 15: There is a point missing after “profile”.

Page 2, line 84: “interambrane” should be replaced by “intermembrane”.

Page 2, line 85: “interference” should be changed for “interfere”.

Page 3, line 142: “eres” should be changed for “were”.

Page 5, line 193: “non-functionnalizeg” should be corrected by “non-functionnalized”.

Page 5, line 211: In the title 2.7, “results” should be corrected for “result”.

Page 5, line 216: It should be “a sample”.

Page 10, line 383: In “Figure 4”, the 4 should be in black and not red. Same point for Figure 7A, page 16, line 590; and Figure 7, page 20, line 679.

Page 11, line 422: The sentence is not correct grammaticaly, please correct it.

Page 13, line 468: “LCLB56” is written “LCLB65”.

Page 14, line 499: “after its” should be changed for “after their”.

Page 16, line 598: “the” should be removed from “of the another example”.

Page 19, line 657: You should correct “functionalize” into “functionalized”.

Page 20, line 730: “therefore it respond” should be “therefore it responds”.

In figure 1, legend is missing informations to understand all parameters that are actually properly defined in the legend of Table 1.

Author Response

Response to Reviewer 3 Comments

Comments of the Reviewer [materials-811885]

We would like to thank the Editor and Reviewer for careful reading, and constructive suggestions for our manuscript that will help us to improve our work. According to this, we have comprehensively revised our manuscript. Hoping that we have addressed all the questions mentioned by the Reviewer, below we included the point-to-point response to each comments of Reviewer.

In manuscript file all of the changes have been provided in red color.

REVIEWER COMMENTS:

The manuscript describes the development, and characterization both physico-chemical and biological of silver nanoparticles functionalized with two different antibiotics. The strategy is not new but results are puzzling in the literature. In this study, the authors performed an in-depth study of their systems. The methods and the processes used are fine and well performed. The results are not spectacular but anyway interesting in the field. After minor corrections the manuscript can be accepted in Materials for me. However, the manuscript has to be read-proof by a native speaker since there is a lot of grammar mistakes. Below are my main comments and minor corrections.

Main comments:

Point 1: Figure 5: The choice of representation for the graphs are not appropriate. Please choose something like histograms. In the current version, we cannot visualize all the data. A clear figure is crucial since this part is tough to understand from the text.

Response 1: Taking into consideration the Reviewer’s suggestion the figure 5 was modified in form of histograms. The authors hope that current form will allowed the reader to follow it.

Point 2: Table 3: Although it seems clear that the concentrations determined for the MIC values in µg/mL correspond to the concentration in Ag or in the antibiotic, but in the case of functionnalized AgNPs, this is unclear. This point has to be clarified and explained how it is calculated since it is crucial to draw proper conclusions concerning these MIC comparisons.

Response 2: We would like to thank the Reviewer for this valuable remark. The  concentrations of functionalized nanoparticles used to determine MIC values has not been explained in the manuscript. Therefore, the authors added an appropriate explanation in section 2.6.1. of "Materials and methods".

Point 3: The same applies for the data presented in Figure 6, in the current form, it is very difficult to conclude about the functionalized AgNPs. The experiments presented in Figure 6 are also not properly designed. Indeed, non-functionalized AgNPs are already highly effective in killing bacteria at 25 µg/mL. It is therefore impossible to see an additive effect in the case of the functionalized AgNPs. This experiment has to be redone at lower concentrations.

Response 3: We would like to thank Reviewer for this valuable remark. However, we would like to point out that investigation of additive effects was not the goal of this work. We focused on the description of the mechanism of functionalization of biologically synthesized silver (bio)nanoparticles in two systems, using two antibiotics from different classes. In addition, we aimed to determine the impact of functionalization on the molecular profile of microorganisms treated with such systems. Moreover, MIC studies indicated no increase in the microbiological efficiency of nanoparticles as a result of functionalization in almost all studied variants. In addition, the MIC value determines the concentration at the level of limit action. The aim of flow cytometry studies was to examine the effective concentrations associated with the application as well as conduction the experiment under the same conditions for all variants. Therefore, a concentration of 25 µg/mL was chosen that was effective in the vast majority of tested cases. The MIC value above this concentration was recorded only in the case of nanoparticles functionalized with ampicillin against S. aureus as well as ampicillin against K. pneumonia and P. aeruginosa which was also confirmed by cytometric analysis. Taking into account the Reviewer's remark, the authors have  improved the manuscript information about the reason for choosing such concentration for flow cytometry tests.

Point 4: The concentration(s) used for the mass spectrometry experiments is not specified in the main text. This is missing as well as the rationale for the choice of this dose.

Response 4: Taking in account the Reviewer’s remark, the missed information about concentrations of stress agents for MALDI experiments, have been added to the manuscript. These doses were selected experimentally to ensure sufficient number of cells for MALDI analysis, with the selected concentration of silver (bio)nanoparticles.

Point 5: Minor corrections.

In the abstract, line 15: There is a point missing after “profile”.

Page 2, line 84: “interambrane” should be replaced by “intermembrane”.

Page 2, line 85: “interference” should be changed for “interfere”.

Page 3, line 142: “eres” should be changed for “were”.

Page 5, line 193: “non-functionnalizeg” should be corrected by “non-functionnalized”.

Page 5, line 211: In the title 2.7, “results” should be corrected for “result”.

Page 5, line 216: It should be “a sample”.

Page 10, line 383: In “Figure 4”, the 4 should be in black and not red. Same point for Figure 7A, page 16, line 590; and Figure 7, page 20, line 679.

Page 11, line 422: The sentence is not correct grammaticaly, please correct it.

Page 13, line 468: “LCLB56” is written “LCLB65”.

Page 14, line 499: “after its” should be changed for “after their”.

Page 16, line 598: “the” should be removed from “of the another example”.

Page 19, line 657: You should correct “functionalize” into “functionalized”.

Page 20, line 730: “therefore it respond” should be “therefore it responds”.

Response 5: We would like to thank the Reviewer for a thorough revision of the manuscript. All suggestions have been taken in consideration and provided to the manuscript.

Point 6: In figure 1, legend is missing informations to understand all parameters that are actually properly defined in the legend of Table 1.

Response 6: According to the Reviewer's comment, Figure 1 was supplemented with the required information suggested by the Reviewer.
